# COMPRESSING GRADIENTS IN DISTRIBUTED SGD BY EXPLOITING THEIR TEMPORAL CORRELATION

## ABSTRACT

We propose SignXOR, a novel compression scheme that exploits temporal correlation of gradients for the purpose of gradient compression. Sign-based schemes such as Scaled-sign and SignSGD (Bernstein et al., 2018; Karimireddy et al., 2019) compress gradients by storing only the sign of gradient entries. These methods, however, ignore temporal correlations between gradients. The equality or non-equality of signs of gradients in two consecutive iterations can be represented by a binary vector, which can be further compressed depending on its entropy. By implementing a rate-distortion encoder we increase the temporal correlation of gradients, lowering entropy and improving compression. We achieve theoretical convergence of SignXOR by employing the two-way error-feedback approach introduced by Zheng et al. (2019). Zheng et al. (2019) show that two-way compression with error-feedback achieves the same asymptotic convergence rate as SGD, although convergence is slower by a constant factor. We strengthen their analysis to show that the rate of convergence of two-way compression with error-feedback asymptotically is the same as that of SGD. As a corollary we prove that two-way SignXOR compression with error-feedback achieves the same asymptotic rate of convergence as SGD. We numerically evaluate our proposed method on the CIFAR-100 and ImageNet datasets and show that SignXOR requires less than $50\%$ of communication traffic compared to sending sign of gradients. To the best of our knowledge we are the first to present a gradient compression scheme that exploits temporal correlation of gradients.

## 1 INTRODUCTION

Distributed optimization has become the norm for training machine learning models on large datasets. With the need to train bigger models on ever-growing datasets, scalability of distributed optimization has become a key focus in the research community. While an obvious solution to growing dataset size is to increase the number of workers, the communication among workers has proven to be a bottleneck. For popular benchmark models such as AlexNet, ResNet and BERT the communication can account for a significant portion of the overall training time (Alistarh et al., 2017; Seide et al., 2014; Lin et al., 2018). The BERT ("Bidirectional Encoder Representing from Transformers") architecture for language models (Devlin et al., 2018) comprises about 340 million parameters. If 32-bit floating-point representation is used one gradient update from a worker amounts to communicating around 1.3GB ($340 \times 10^6$ parameters $\times$ 32 bits per parameter $\times 2^{-33}$ gigabytes per bit $\approx$ 1.3GB). Frequently communicating such large payloads can easily overwhelm the network resulting in prolonged training times. In addition, large payloads may increase other forms of costs in distributed optimization. Novel approaches such as federated learning employ mobile devices as worker nodes. Exchanging information with mobile devices is heavily constrained due to communication bandwidth and budget limitations. Therefore, communication remains an important bottleneck in distributed optimization and reducing communication is of utmost importance.

Gradient compression alleviates the communication bottleneck. The idea is to apply a compression scheme on gradients before sending them over the network. There has been an increasing amount of literature on gradient compression within the last few years (Seide et al., 2014; Aji & Heafield, 2017; Alistarh et al., 2017; Wen et al., 2017; Wangni et al., 2018; Wu et al., 2018; Lin et al., 2018; Wang et al., 2018). Such compression schemes have been demonstrated to work well with distributed stochastic gradient descent (SGD) and its variants. However, SGD with arbitrary compres-

sion schemes may not converge. Karimireddy et al. (2019) give one example of non-convergence. The recently proposed error-feedback based algorithms (Stich et al., 2018; Karimireddy et al., 2019) circumvent the convergence issue. Error-feedback methods accumulate the compression error and feed it back to the input of the compression scheme so that the error gets transmitted over subsequent iterations. The dist-EF-SGD algorithm proposed by Zheng et al. (2019) applies error-feedback to two-way compression, in which both worker-to-master and master-to-worker communications are compressed. Theoretical guarantees provided by dist-EF-SGD are valid for all compression schemes that fall under the definition of '$\delta$-approximate compressors', also referred to as $\delta$-compressors. The authors prove that error-feedback with two-way compression asymptotically achieves the $\mathcal{O}(1/\sqrt{T})$ convergence rate of SGD. However, the analysis by Zheng et al. (2019) suggests that dist-EF-SGD converges slower than SGD by a constant factor.

Our contributions in this paper are as follows. We propose SignXOR, a novel compression scheme that exploits temporal correlation of gradients. We prove that SignXOR is a $\delta$-compressor, and we provide convergence guarantees for SignXOR by employing dist-EF-SGD. We strengthen the convergence bound by Zheng et al. (2019) to show that dist-EF-SGD asymptotically converges at the same $\mathcal{O}(1/\sqrt{T})$ rate as SGD. Consequently, we show that the proposed method asymptotically achieves the SGD convergence rate. We empirically validate the proposed method on CIFAR-100 and ImageNet datasets and demonstrate that the ratio between total communication budgets of SignXOR and Scaled-sign is less than $50\%$.

**Notation:** For $x \in \mathbb{R}^d$, $x[j]$ denotes the $j$th entry of $x$, $\|x\|_1$ denotes the $\ell_1$-norm, and $\|x\|$ denotes the $\ell_2$-norm. For vector inputs $\mathrm{sgn}(\cdot)$ function outputs the sign of the input element-wise. The index set $\{1, \ldots, n\}$ is denoted by $[n]$, and $\odot$ denotes elementwise multiplication.

## 2 RELATED WORK

The most common gradient compression schemes can be categorized into those based on sparsification and those based on quantization. The methods based on sparsification such as Top-$k$, Rand-$k$ (Stich et al., 2018; Lin et al., 2018) and Spectral-ATOMO (Wang et al., 2018) preserve only the most significant gradient elements, effectively reducing the quantity of information carrying gradient components. On the other hand, the methods based on quantization such as QSGD (Alistarh et al., 2017), TernGrad (Wen et al., 2017) and SignSGD (Bernstein et al., 2018) reduce the overall floating-point precision of the gradient. Therefore, these two classes of methods can be respectively thought of as approaches that reduce the quantity versus the quality of the gradient. One can think of this in analogy to image compression. For example, JPEG image compression that is based on discrete cosine transform determines both which transform coefficients to store (the quantity) and at what level of resolution to store those coefficients (the quality).

Sign-based compression schemes such as Scaled-sign, SignSGD and Signum (Bernstein et al., 2018) sit at the far end of quantization-based algorithms. Such schemes quantize real values to only two levels, $+1$ and $-1$. For example, the compressing function of Scaled-sign takes in a vector $x \in \mathbb{R}^d$, and the decompressing function outputs the vector $(\|x\|_1/d) \, \mathrm{sgn}(x)$. This means that the compressed representation needs to store only the sign of each entry $x[j]$, along with the scaling constant $\|x\|_1/d$. In practice one can avoid the 'zero' output of $\mathrm{sgn}$ by mapping it to $+1$ or $-1$. This allows the two outcomes $+1$ and $-1$ to be represented using one bit per entry, making the size of the compressed representation $d + 32$ bits in total (assuming 32-bit single-precision representation of the scaling constant). As per Shannon's source coding theorem (MacKay, 2003, p. 81), the sequence of $+1$ and $-1$ can be further compressed without any information loss if the probability of encountering $+1$ is different from that for $-1$. However, in our experiments on Scaled-sign compression we observe that both outputs are equally likely across all iterations.

Any lossy gradient compression scheme introduces noise, also known as distortion, in addition to the measurement noise that is already present in the stochastic gradients computed by the workers. It is reasonable to expect the additional compression error hurts the convergence rate of the algorithm. However, it has been empirically observed that significant compression ratios can be achieved before observing any impact on convergence (Seide et al., 2014; Alistarh et al., 2017). One can achieve even greater compression while keeping the convergence rate nearly the same by employing error-feedback (Stich et al., 2018; Karimireddy et al., 2019; Zheng et al., 2019). Algorithms based on error-feedback accumulate the compression error in past iterations and add it to the input of the

compressing function. This allows all of the gradient information to get transmitted over a sequence of iterations, albeit with a delay. For smooth functions the gradient does not change considerably, therefore, the delay does not significantly impact the rate of convergence.

The dist-EF-SGD algorithm proposed by Zheng et al. (2019) is based on error-feedback and offers two-way compression under the master-worker topology. The dist-EF-SGD algorithm provides convergence guarantees for $\delta$-compressors. An operator $C : \mathbb{R}^d \to \mathbb{R}^d$ is a $\delta$-compressor for all $x \in \mathbb{R}^d$ if $\|C(x) - x\|^2 \le (1 - \delta)\|x\|^2$ for some $\delta \in (0, 1]$. The $\delta$ is a measure of the distortion due to applying $C$. A good compressor will have $\delta$ close to $1$ and a large compression ratio. One can show that Scaled-sign is a $\frac{1}{d}$-approximate compressor. The Scaled-sign compression scheme with dist-EF-SGD offers convergence guarantees under standard assumptions even though Scaled-sign without error-feedback may not converge (Karimireddy et al., 2019). Zheng et al. (2019) generalize the dist-EF-SGD algorithm to include blockwise compression, in which the gradient is partitioned into blocks that are compressed separately. A natural partitioning method is to consider as blocks the elements of a deep neural network such as tensors, matrices and vectors. Blockwise compression allows better exploitation of redundancy that may be present only within a block.

Zheng et al. (2019) empirically demonstrate that Scaled-sign with dist-EF-SGD achieves almost the same performance of SGD with respect to training loss and test accuracy. This indicates that we can allow more distortion in the compression before observing a significant impact on training performance. The proposed SignXOR compression is based on allowing additional distortion in the interest of achieving a higher compression ratio.

## 3 PROPOSED ALGORITHM: SIGNXOR

Our proposal is motivated by the concept of delta encoding. Delta encoding refers to techniques that store data as the difference between successive samples, rather than directly storing the samples themselves (Smith et al., 1997). One often encounters delta encoding in applications such as integer compression and video compression. For example, it is more space efficient to store the first-order differences of a digitized audio signal than to store the values of the original signal. First-order differences have smaller magnitudes compared to the original sequence and, therefore, the differences can be represented by a comparatively smaller number of bits. A similar approach is used in the inter-picture prediction method in high efficiency video coding (HEVC) (Sze et al., 2014). Inter-picture prediction makes use of temporal correlations across video frames encoding the differences between frames. This requires less storage compared to storing each video frame. Jiang et al. (2018) employ delta encoding in a more related application to distributed optimization. At the core of their algorithm, delta encoding is employed to compress an increasing sequence of integers.

Our SignXOR algorithm applies delta encoding to represent temporal changes in the gradient. In essence, SignXOR maintains a binary vector that indicates whether the sign of a gradient entry is equal (or not equal) to the sign of the corresponding entry in the previous gradient. The equality (or non-equality) can be represented by a binary $1$ (or $0$). This procedure resembles the binary XOR operation, hence the name SignXOR. We employ a generalized version of original dist-EF-SGD by Zheng et al. (2019) (the original dist-EF-SGD is specified in Algorithm 2 therein) to provide convergence guarantees for the proposed compression scheme. The generalization is to make dist-EF-SGD compatible with SignXOR. We outline the proposed method in Algorithm 1 and Algorithm 2.

**Generalized dist-EF-SGD:** Algorithm 1 presents generalized dist-EF-SGD for setup with a master and $n$ workers. The three main differences between the generalized and original dist-EF-SGD versions are as follows. First, Algorithm 1 delegates compression and decompression tasks to two separate functions *encode* and *decode*. Second, Algorithm 1 maintains a vector $\bar{g}_k$ at all nodes including the master. The vector $\bar{g}_k$ is the average gradient all workers used to update the parameter vector $w_k$ in the last iteration. Third, the *encode* and *decode* functions each take the last gradient $\bar{g}_k$ as the second argument. This is in contrast to the original dist-EF-SGD algorithm in which compression is based only on the gradient in the current iteration. Since there are no differences related to compression performance between the original and generalized dist-EF-SGD algorithms, they encompass the same theoretical guarantees.

The compressing function *encode* takes in two inputs and outputs $\mathcal{G}_k^i$. This output is the actual payload sent to master over the communication channel. In addition to $\mathcal{G}_k^i$, the $i$th worker also

---

**Algorithm 1:** Generalized dist-EF-SGD compatible with SignXOR

    **input**    : initial parameter vector $w_0$; step sizes $\{\eta_0, \ldots, \eta_{T-1}\}$
    **initialize:** let $\mathbf{0}$ be the all-zeros vector of dimension $d$ ;
                 let $\bar{g}_0 \in \mathbb{R}^d$ be a vector with entries sampled uniformly from $[-1, 1]$;
                 **on $i$th worker** store $\bar{g}_0$ ; set $e_0^i$ to $\mathbf{0}$ ; set parameter vector to $w_0$ ;
                 **on master** store $\bar{g}_0$ ; set $e_0$ to $\mathbf{0}$ ;

1 **for** $k \in \{0, \ldots, T-1\}$ **do**
2      **on $i$th worker**
3          compute $\hat{g}_k^i = g_k^i + \frac{1}{\eta_k} e_k^i$ where $g_k^i$ is the stochastic gradient at $w_k$ ;
4          compute $\mathcal{G}_k^i = encode(\hat{g}_k^i, \bar{g}_k)$ and $\bar{g}_k^i = decode(\mathcal{G}_k^i, \bar{g}_k)$ ;
5          send $\mathcal{G}_k^i$ to master and update error $e_{k+1}^i = \eta_k(\hat{g}_k^i - \bar{g}_k^i)$ ;
6          receive $\mathcal{G}_k$ from master and compute $\bar{g}_{k+1} = decode(\mathcal{G}_k, \bar{g}_k)$ ;
7          update parameter vector $w_{k+1} = w_k - \eta_k \bar{g}_{k+1}$ ;

8      **on master**
9          receive $\mathcal{G}_k^i$ and compute $\bar{g}_k^i = decode(\mathcal{G}_k^i, \bar{g}_k)$ for all $i \in [n]$ ;
10          compute $\hat{g}_k = \frac{1}{n} \sum_{i \in [n]} \bar{g}_k^i + \frac{1}{\eta_k} e_k$ ;
11          compute $\mathcal{G}_k = encode(\hat{g}_k, \bar{g}_k)$ and $\bar{g}_{k+1} = decode(\mathcal{G}_k, \bar{g}_k)$ ;
12          broadcast $\mathcal{G}_k$ to all workers and update $e_{k+1} = \eta_k(\hat{g}_k - \bar{g}_{k+1})$ ;

---

computes $\bar{g}_k^i$ which is what the master will obtain by decompressing $\mathcal{G}_k^i$. The vector $\bar{g}_k^i$ is used to update $e_{k+1}^i$, the compression error fed back in the next iteration. The master collects $\mathcal{G}_k^i$ from all workers and decompresses each to obtain $\bar{g}_k^i$. All workers receive the master broadcast $\mathcal{G}_k$, and input it, along with $\bar{g}_k$, to the *decode* function to decompress and obtain $\bar{g}_{k+1}$. Note that in the $k$th iteration all nodes use the same $\bar{g}_k$ vector as the second argument in the *encode* and *decode* functions.

The *encode* and *decode* functions corresponding to SignXOR compression scheme are specified in Algorithm 2. For the ease of explanation we consider the specific case when the master compresses $\hat{g}_k$ to obtain $\mathcal{G}_k$, and the workers decompress $\mathcal{G}_k$ to obtain $\bar{g}_{k+1}$ as an approximation to $\hat{g}_k$.

---

**Algorithm 2:** SignXOR compression and decompression

    **input:** hyperparameter $0 \leq \alpha < \frac{1}{4}\left(1 - \sqrt{1 - \frac{1}{d}}\right)^2 < 1$ ;
1 **function** $encode(x \in \mathbb{R}^d, y \in \mathbb{R}^d)$**:**
2      compute $r$, the fraction of $+1$'s in $\mathrm{sgn}(x)$ ;
3      compute $q$, the fraction of elements in $x$ such that $\mathrm{sgn}(x[j]) = \mathrm{sgn}(y[j])$ ;
4      initialize binary vector $b \in \{0, 1\}^d$ to all-zeros ;
5      for all $j \in [d]$, if $\mathrm{sgn}(x[j]) = \mathrm{sgn}(y[j])$ set $b[j] = 1$ with probability $1 - \alpha$ ;
6      compute $p$, the fraction of $1$'s in $b$ ;
7      compress $b$ with a lossless scheme and compute scalar $a = \|x\|_1 / d$ ;
8      **output** $\mathcal{G} = \{a, \text{compressed representation of } b\}$ ;

9 **function** $decode(\mathcal{G} = \{a \in \mathbb{R}, \text{compressed representation of a vector } b \in \{0, 1\}^d\}, y \in \mathbb{R}^d)$**:**
10      expand and decompress $\mathcal{G}$ to obtain $a$ and $b$ ;
11      **output** $a\,\mathrm{sgn}(y) \odot (2b - 1)$ ;

---

**Compressing function:** We consider the case when master calls *encode* with arguments $x = \hat{g}_k$ and $y = \bar{g}_k$. Note that the scalars $r$, $q$ and $p$ are not used anywhere in Algorithm 2. These scalars help us describe the algorithm and also become useful in our theoretical analysis. The output of *encode* function is random when $\alpha \neq 0$, and deterministic when $\alpha = 0$. Let us first consider $\alpha = 0$, in which case $b[j] = 1$ if and only if $\mathrm{sgn}(\hat{g}_k[j]) = \mathrm{sgn}(\bar{g}_k[j])$. This implies that $p$ is the fraction of entries in $\hat{g}_k$ and $\bar{g}_k$ that have the same signs. This is a measure of the (positive or negative) correlation between

the two vectors. The core idea in the proposed compression scheme is to compress the binary vector $b$ using a lossless compression scheme. Shannon's source coding theorem (MacKay, 2003, p. 81) states that $d$ i.i.d. random variables each with entropy $H(p) = -p \log_2 p - (1 - p) \log_2(1 - p)$ can be compressed into approximately $dH(p)$ bits with negligible risk of information loss, as $d \to \infty$. In our case, while the length of the parameter vector $d$ is well over a million for models of practical interest, the entries of $b$ are not necessarily i.i.d.. However, we demonstrate in Section 5 that there exist readily available lossless compression algorithms that can compress $b$ to very close to $dH(p)$ bits, the Shannon limit. Note that binary entropy $H(p)$ is symmetric around $p = 0.5$, and satisfies $0 \leq H(p) \leq 1$ with $H(0.5) = 1$. When $p \approx 0.5$ the size of the compressed representation gets close to $d$, which is same as that offered by Scaled-sign. Further compression of $b$ is only possible if $p$ is away from 0.5. In our experiments presented in Section 5.2 we observe that when $\alpha = 0$, $p$ remain close to but slightly lower than 0.5. This implies a low correlation between $\hat{g}_k$ and $\bar{g}_k$. We remedy this issue by making $\alpha > 0$. Next we explain how making $\alpha > 0$ yields compression gains and, at the same time, induces correlation between $\hat{g}_k$ and $\bar{g}_k$.

First, note that increasing $\alpha$ introduces distortion to $b$, driving $p$ away from 0.5 and towards 0. A lower $p$ decreases the entropy of $b$, yielding a higher compression ratio realized with the lossless compressor. Since we start with a $p$ slightly less than 0.5, if we were to drive $p$ towards 1 we would first increase $H(p)$ before starting to incur compression gains. For this reason, we design the encoder to push $p$ towards zero. In summary, increasing $\alpha$ offers rate savings in the current iteration by adding distortion to the current gradient.

Second, we explain how added distortion induces correlation between $\hat{g}_k$ and $\bar{g}_k$. Note that $q$ is a measure of correlation between the two vectors $\mathrm{sgn}(\hat{g}_k)$ and $\mathrm{sgn}(\bar{g}_k)$ measured *prior* to adding distortion using $\alpha$. We emphasize that $q$ does not depend on the errors introduced in the *current* iteration. Rather, $q$ depends on $\alpha$ only through the *past* iterations. In the experimental results presented in Section 5.2 we observe that increasing $\alpha$ also decreases $q$. This means that in a given iteration the inputs to the encoder are already correlated. Therefore, the $b$ vector that encodes equality (or non-equality) of $\mathrm{sgn}(\hat{g}_k)$ and $\mathrm{sgn}(\bar{g}_k)$ can be compressed even without adding distortion in the current iteration. The underlying mechanism that causes this temporal correlation in our encoder is error-feedback. Recall that the idea behind error-feedback is to keep track of the compression error in the last iteration and add it back to the input of the encoder in the current iteration (with correction for the step size). Specifically, $\hat{g}_k$ includes the compression error incurred in $\bar{g}_k$. This feedback system induces the temporal relation between the two vectors that we see through $q$. In summary, our compression mechanism interacts with the error-feedback mechanism to increase temporal correlation which we then exploit to realize further compressive gains.

The suggested upper bound for $\alpha$ ensures theoretical convergence of the SignXOR algorithm. We show in Section 5 that in practice $\alpha$ can be increased considerably more than the suggested upper bound before seeing an impact on the training performance.

**Decompressing function:** Let us now consider the case when a worker calls *decode* with arguments $\mathcal{G} = \mathcal{G}_k$ and $y = \bar{g}_k$. The first argument $\mathcal{G}_k$ contains the scalar $a$ and the compressed representation of binary vector $b$, which can be recovered exactly as compression of $b$ is lossless. Noting that $b$ is computed with $\bar{g}_k$ as the second argument to the *encode* function, the output of the decoder is obtained by inverting the sign of $\bar{g}_k[j]$ whenever $b[j]$ is 0, and by scaling the result by $a$. One can compactly express the output of this operation as $a \, \mathrm{sgn}(\bar{g}_k) \odot (2b - 1)$.

**Remarks:** Note that we can recover the Scaled-sign compression scheme by setting $\alpha = 0$ in Algorithm 2. In comparison to SignXOR, the compressed representation of Scaled-sign stores the sign of each entry in $\hat{g}_k$, and the scaling constant $a = \|\hat{g}_k\|_1 / d$. We demonstrate in Section 5.2 that $r$, the fraction of $+1$'s in $\mathrm{sgn}(\hat{g}_k)$, is approximately 0.5 across all iterations. This means that we cannot compress further the sequence of signs, and the encoded representation of Scaled-sign requires at least $d$ bits. Algorithm 1 can be easily extended to accommodate blockwise compression as was explained by Zheng et al. (2019). In blockwise compression the gradient $\hat{g}_k$ is partitioned into blocks, and the blocks are processed separately using *encode* and *decode* functions. In our numerical experiments we employ the blockwise extension of Algorithm 1.

## 4 THEORETICAL GUARANTEES

In this section we summarize our theoretical results on Algorithm 1 and Algorithm 2. First, we prove that the SignXOR compression scheme presented in Algorithm 2 is a $\delta$-compressor. Second, we show that for any $\delta$-compressor the generalized dist-EF-SGD scheme in Algorithm 1 converges at the same $\mathcal{O}(1/\sqrt{T})$ rate as SGD. Putting these two together yields the desired result.

**SignXOR is a $\delta$-compressor:** We consider the general form of *encode* and *decode* functions, i.e., with inputs $(x, y)$ for *encode*, and with inputs $(\mathcal{G}, y)$ for *decode*. Let us define the operator $C_\alpha^y$ : $\mathbb{R}^d \to \mathbb{R}^d$ where $C_\alpha^y(x) = a \operatorname{sgn}(y) \odot (2b - 1)$ with $a = \frac{\|x\|_1}{d}$ and the binary vector $b$ is such that $b = 1$ with probability $1 - \alpha$ if $\operatorname{sgn}(x) = \operatorname{sgn}(y)$, and $b = 0$ otherwise. Note that $a$ and $b$ have the same meaning as in Algorithm 2. The operator $C_\alpha^y(x)$ is representative of $decode(encode(x, y), y)$, therefore, $C_\alpha^y(x)$ is the SignXOR compressor. Since $C_\alpha^y$ is a randomized operator, we show that it is a $\delta$-compressor *in expectation* as stated in Theorem 1.

**Theorem 1.** *There exist a $\delta \in (0, 1]$ such that $\mathbb{E}[\|C_\alpha^y(x) - x\|^2] \leq (1 - \delta)\|x\|^2$ for all $y \in \mathbb{R}^d$ if $\alpha < \frac{1}{4}\left(1 - \sqrt{1 - \frac{1}{d}}\right)^2$.*

The proof is provided in Appendix A.1. Although the suggested upper bound for $\alpha$ is extremely small for large $d$, we demonstrate in Section 5 that in practice $\alpha$ can be set quite close to 1. Next we discuss the convergence rate of dist-EF-SGD for an arbitrary $\delta$-compressor.

**Convergence rate of dist-EF-SGD:** Zheng et al. (2019) compare the convergence rates of dist-EF-SGD and vanilla SGD in their Corollary 1. As authors note, although the convergence rates of the two algorithms are same in $\mathcal{O}$ notation, the former is slower by a constant factor. The differences between the bounds for dist-EF-SGD and vanilla SGD in Corollary 1 are as follows. In both cases, for large $T$ the dominant terms are those with $\sqrt{T}$ in denominator. The ratio between the dominant terms corresponding to dist-EF-SGD and vanilla SGD is 1.5, suggesting that the former is always slower than the latter by a factor of 1.5. In our Theorem 2 we strengthen the bound for dist-EF-SGD so that the dominant terms in the two algorithms are the same. This means that dist-EF-SGD achieves same convergence rate as vanilla SGD in the limit as $T \to \infty$. Our proof is for an arbitrary $\delta$-compressor that is not necessarily SignXOR. To this end we consider Algorithm 1 and define $C(x) = decode(encode(x, y), y)$. We assume that $C$ is a $\delta$-compressor for all $y \in \mathbb{R}^d$, i.e., $\mathbb{E}[\|C(x) - x\|^2] \leq (1 - \delta)\|x\|^2$ for some $\delta \in (0, 1]$. We setup the optimization problem next.

Let $F : \mathbb{R}^d \to \mathbb{R}$ be a function that is lower bounded by $F_*$ and has a gradient $\nabla F$. We consider a distributed optimization setup with a master and $n$ workers. For some $w_k \in \mathbb{R}^d$ the $i$th worker calculates the stochastic gradient $g_k^i$ at $w_k$. We assume that $g_k^i$ is an unbiased estimate of $\nabla F(w_k)$, and that $g_k^i$ has bounded variance. Specifically, we assume that $g_k^i$ satisfies the two properties $\mathbb{E}[g_k^i|w_k] = \nabla F(w_k)$ and $\mathbb{E}[\|g_k^i - \nabla F(w_k)\|^2] \leq \sigma^2$. We also assume that $F$ is $L$-Lipschitz smooth, and that the gradient of $F$ is bounded. The latter implies that $\mathbb{E}[\|g_k^i\|^2] \leq G^2$ for some scalar $G$. The master and workers generate a sequence $\{w_0, \ldots, w_T\}$ as per Algorithm 1. Convergence results of this system is summarized in Theorem 2.

**Theorem 2.** *For a given $w_0$ and a step size schedule $\eta_k = \frac{1}{L\sqrt{T}}\left(1 - \frac{1}{2T^{1/4}}\right)$, the convergence of the system outlined in Algorithm 1 after $T$ iterations is given by*

$$\mathbb{E}\left[\min_{k=0,\ldots,T-1} \|\nabla F(w_k)\|^2\right] < \frac{2L(F(w_0) - F_*) + \frac{\sigma^2}{n}}{2\sqrt{T} - 1}$$
$$+ \frac{2L(F(w_0) - F_*) + \frac{8(1-\delta)G^2}{\delta^2}\left(1 + \frac{1}{\delta^2}\right)}{2T^{3/4} - T^{1/4}} + \mathcal{O}\left(\frac{1}{T}\right). \quad (1)$$

We defer the proof of Theorem 2 to Section A.2 in the Appendix. In comparison, the bound for SGD has only the first term in (1). Note that the last two terms in (1) converge to zero faster than the first term. This means that dist-EF-SGD asymptotically achieves the same convergence rate as SGD. Since SignXOR is a $\delta$-compressor we conclude that the proposed algorithm converges asymptotically at the same rate as SGD.

Table 1: Details of experiments and summary of findings.

| Dataset | Model | Parameters | Workers | Batch size | $B_X/B_S$ |
|---------|-------|-----------|---------|-----------|-----------|
| CIFAR-100 | ResNet-18 | 12,596,388 | 4 | 32 | 42% |
| ImageNet-32 | WRN-28-2 | 1,595,320 | 4 | 64 | 43% |
| ImageNet | ResNet-50 | 25,583,592 | 16 | 16 | 47% |

## 5 NUMERICAL EXPERIMENTS

We perform multi-GPU experiments on three datasets: CIFAR-100, ImageNet-32 and ImageNet. The ImageNet-32 dataset (Chrabaszcz et al., 2017) is a down-sampled version of the ImageNet dataset in which the images are of size $32 \times 32$ pixels. All other properties such as the number of images and the number of classes are the same as the original ImageNet dataset. We experiment with ImageNet-32 as an alternative to ImageNet due to the latter's heavy consumption of hardware resources. The additional experiments presented in Section 5.2 are performed with ImageNet-32. We train a wide residual network classifier (Zagoruyko & Komodakis, 2016) of depth 28 and width 2 (WRN-28-2) using the ImageNet-32 dataset. The CIFAR-100 and ImageNet datasets are used to train the ResNet-18 and ResNet-50 (He et al., 2016) models respectively. In all cases the dataset is partitioned to equal sizes and distributed among the workers.

We implement all algorithms using the TensorFlow and Horovod (Sergeev & Balso, 2018) packages. Since the communications component in Horovod is designed for a master-less setup, we simulate a master-worker environment in our implementation. We choose to use the Lempel–Ziv–Markov chain algorithm (LZMA) as the lossless scheme that compresses the binary vector $b$. The LZMA compression scheme is a member of the Lempel–Ziv family of compression algorithms (MacKay, 2003, p. 119). While there exist other compression schemes in which for large $d$ the compression ratio converge to $H(p)$ much faster than the Lempel–Ziv algorithms, in our experiments we employ LZMA as its implementation is readily available in Python. We refer the reader to Appendix Section A.3 for a comparison of $H(p)$ and the compression ratio of LZMA. In our implementation the *encode* and *decode* functions are not tuned to achieve best computational performance. We, therefore, do not compare or plot measurements with respect to time. In our experimental setup, a single machine hosts four workers and each worker runs on a dedicated Tesla P100 16GB GPU.

We compare the performance of SignXOR to SGD and dist-EF-SGD with Scaled-sign compression. For Scaled-sign we use the original dist-EF-SGD scheme as the error-feedback mechanism. For SignXOR we use the generalized dist-EF-SGD outlined in Algorithm 1. Note that there are no compression performance related differences between the original and the generalized versions of dist-EF-SGD. In both cases we use blockwise compression, in which the gradients corresponding to tensors, matrices and vectors are compressed and decompressed separately. Training is started with a learning rate of $0.1$, and the learning rate is multiplied iteratively by $0.1$ at certain epochs. These epochs are multiples of 8, 10 and 5 for the experiments involving the CIFAR-100, ImageNet-32, and ImageNet datasets, respectively. We use an $\ell_2$ weight regularizer in all experiments. The regularizer is scaled by $10^{-3}$ in CIFAR-100 experiments, and by $10^{-4}$ in the other experiments. Table 1 summarizes some additional details of the experiments. Batch size is per worker.

We define the following measurement to help us interpret our experimental results. Recall that in Algorithm 1 $\mathcal{G}_k^i$ and $\mathcal{G}_k$ are the payloads that workers send and receive. Let $s_k$ and $s_k^i$ denote the number of bits required to store $\mathcal{G}_k$ and $\mathcal{G}_k^i$ in bits. Over $T$ iterations with SignXOR, each worker on average sends and receives a total of $B_X = \frac{1}{n} \sum_{k=0}^{T-1} \sum_{i \in [n]} (s_k^i + s_k)$ bits. In contrast, if Scaled-sign compression is used each worker communicates $d$ bits in each iteration, making the total number of bits sent and received by each worker $B_S = 2Td$. Therefore, $B_X/B_S$ is the ratio between the total bit usage of each algorithm. We tune the hyperparameter $\alpha$ in Algorithm 2 so that SignXOR achieves the same test accuracy as Scaled-sign within the same number of epochs.

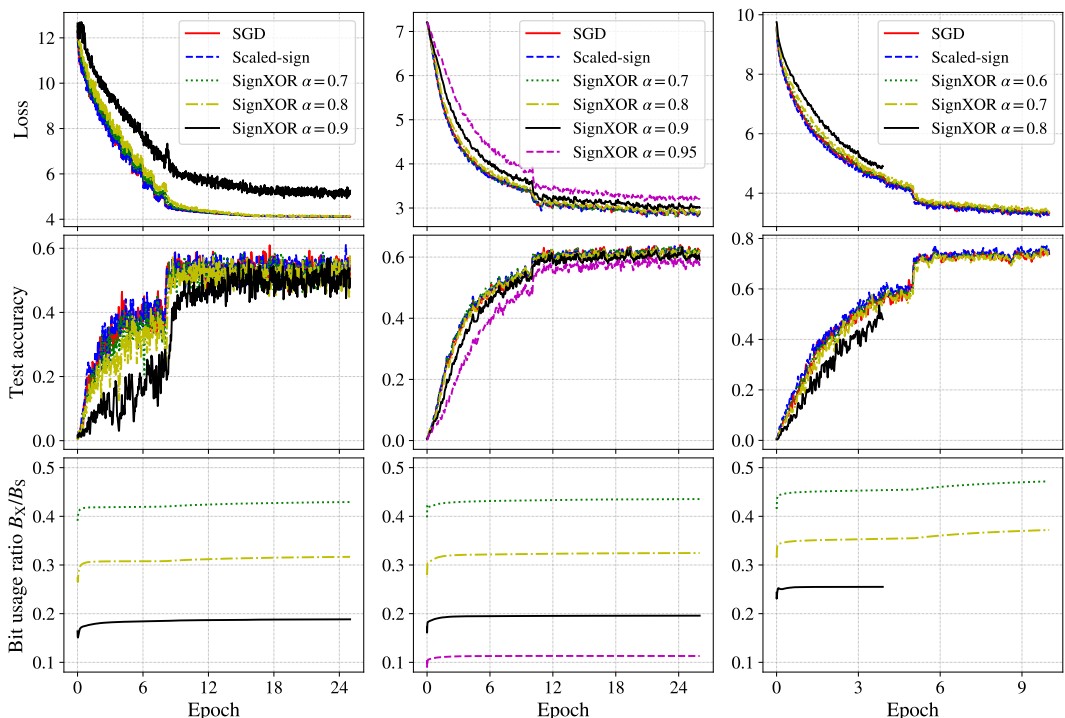

Figure 1: The three columns present experimental results for CIFAR-100 (left), ImageNet-32 (middle), ImageNet (right). In each case the three rows present plots of training loss (top), test accuracy (middle) and $B_X/B_S$ (bottom). The test accuracy plots indicate top-1 test accuracy for CIFAR-100, and top-5 for the other two. We run SignXOR experiments for multiple $\alpha$ values.

## 5.1 MAIN RESULTS

Figure 1 presents results from multiple runs of the experiment with different values for $\alpha$. For the CIFAR-100 dataset, the SignXOR test accuracy plot corresponding to $\alpha = 0.7$ closely follows test accuracy of Scaled-sign and SGD. The same observation is true for plots corresponding to $\alpha = 0.7$ and $\alpha = 0.6$ in ImageNet-32 and ImageNet experiments respectively. For these choices of $\alpha$, the plots of $B_X/B_S$ indicate that in each case SignXOR achieves the top test accuracy with less than $50\%$ of communication required by Scaled-sign. The last column in Table 1 summarizes the lowest $B_X/B_S$ that SignXOR achieves while attaining the same test accuracy as Scaled-sign by the end of training. Figure 1 also indicates that we can make $\alpha$ slightly larger than the choices listed in Table 1 to achieve an even smaller $B_X/B_S$ with some reduction in the test accuracy. For example, the ImageNet-32 experiment corresponding to $\alpha = 0.9$ achieves within $0.3$ test accuracy of Scaled-sign, but only using $12\%$ of the communications usage of Scaled-sign.

## 5.2 DETAILS OF SIGNXOR RESULTS

Figure 2 presents the evolution of $r$, $q$ and $p$ in Algorithm 2 for experiments related to ImageNet-32 dataset. The left-sub figure in Figure 2 demonstrates that $r$, the fraction of $+1$'s in $\mathrm{sgn}(\hat{g}_k)$ is approximately $0.5$ across all iterations. Consequently, the sequence of signs in $\mathrm{sgn}(\hat{g}_k)$ cannot be further compressed. Note that this observation is true for all $\alpha$. Since $\alpha = 0$ recovers the Scaled-sign compression, we conclude that the output of Scaled-sign cannot be further compressed using a lossless compression scheme. The middle-sub figure in Figure 2 plots $q$, the fraction of entries in $\hat{g}_k$ such that $\mathrm{sgn}(\hat{g}_k[j]) = \mathrm{sgn}(\bar{g}_k[j])$. Note that $q$ remains close to $0.5$ when $\alpha = 0$, and goes down as $\alpha$ increases. Recall that $q$ is computed prior to adding distortion using $\alpha$. This implies that the naturally present correlation between the two vectors $\mathrm{sgn}(\hat{g}_k)$ and $\mathrm{sgn}(\bar{g}_k)$, i.e. temporal correlation, increases with $\alpha$. The right-sub figure includes plots of $p$, the fraction of $1$'s in $b$. Note that as per Algorithm 2 we have the relationship $\mathbb{E}[p] = q(1 - \alpha)$.

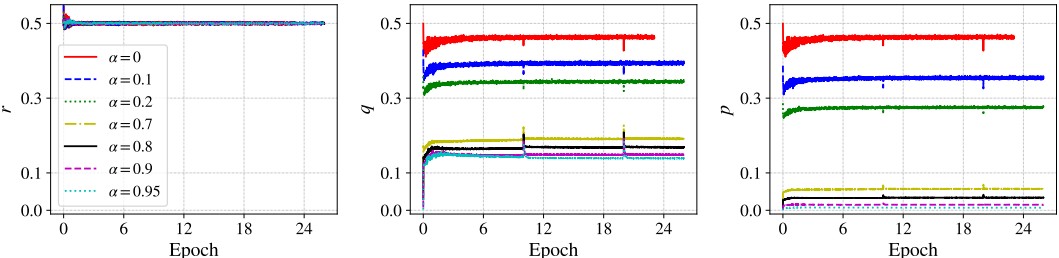

Figure 2: Evolution of $r$, $q$ and $p$ in Algorithm 2 for experiments conducted using the ImageNet32 dataset. Values are collected when the master calls the *encode* function with arguments $x = \hat{g}_k$ and $y = \bar{g}_k$. The peaks at epochs 10 and 20 are due to the drop in the learning rate at these epochs.

## 6 CONCLUSIONS AND FUTURE WORK

In this paper we propose SignXOR, a novel compression algorithm that exploits the temporal correlation of consecutive gradients in SGD. The proposed algorithm builds off Scaled-sign (Bernstein et al., 2018; Karimireddy et al., 2019) and dist-EF-SGD (Zheng et al., 2019). The SignXOR algorithm combines amplitude quantization with the exploitation of temporal correlation. We accomplish this in two stages. The first employs Scaled-sign which quantizes gradient entries into two levels, and the second compresses by exploiting temporal correlation of quantized gradients. It is interesting to investigate how the second stage can be improved to accommodate more elaborate compression mechanisms beyond a two-level quantizer, such as QSGD (Alistarh et al., 2017), Top-$k$ (Stich et al., 2018; Lin et al., 2018) and Spectral-ATOMO (Wang et al., 2018). Also, we conjecture that one must be able to compress better by jointly considering the two stages. This way, one can exploit in the second stage some of the information embedded in the gradient that may have been removed in the first stage. We leave these two tasks as future work.

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

# A    APPENDIX

## A.1    PROOF OF THEOREM 1

*Proof.* We have

$$
\begin{aligned}
\|C_\alpha^y(x) - x\| &= \|a\,\mathrm{sgn}(y) \odot (2b - 1) - x\| \\
&= \|a\,\mathrm{sgn}(y) \odot (2b - 1) - a\,\mathrm{sgn}(x) + a\,\mathrm{sgn}(x) - x\| \\
&\leq a\|\mathrm{sgn}(y) \odot (2b - 1) - \mathrm{sgn}(x)\| + \|a\,\mathrm{sgn}(x) - x\|,
\end{aligned}
$$

where the last step follows from the subadditivity of $\ell_2$-norm. Squaring both sides and taking the expectation with respect to the randomness in $C_\alpha^y$ yields

$$
\begin{aligned}
\mathbb{E}[\|C_\alpha^y(x) - x\|^2] \leq{}& \mathbb{E}[a^2\|\mathrm{sgn}(y) \odot (2b - 1) - \mathrm{sgn}(x)\|^2] + \|a\,\mathrm{sgn}(x) - x\|^2 \\
&+ 2\mathbb{E}[a\|\mathrm{sgn}(y) \odot (2b - 1) - \mathrm{sgn}(x)\|]\|a\,\mathrm{sgn}(x) - x\|
\end{aligned} \tag{2}
$$

Next we bound each term in (2). In the following we use the property $\|x\| \leq \|x\|_1 \leq \sqrt{d}\|x\|$, which shows that $\|x\|_1$ is both lower and upper bounded by $\|x\|$ to within a constant factor.

For the second term in (2) we have

$$
\begin{aligned}
\|a\,\mathrm{sgn}(x) - x\|^2 &= \left\| \frac{\|x\|_1}{d}\,\mathrm{sgn}(x) - x \right\|^2 \\
&= \left\| \frac{\|x\|_1}{d}\,\mathrm{sgn}(x) \right\|^2 - 2\left\langle \frac{\|x\|_1}{d}\,\mathrm{sgn}(x), x \right\rangle + \|x\|^2 \\
&= \frac{\|x\|_1^2}{d^2}\|\mathrm{sgn}(x)\|^2 - 2\frac{\|x\|_1}{d}\langle \mathrm{sgn}(x), x \rangle + \|x\|^2,
\end{aligned}
$$

which can be simplified and upper bounded as

$$
\|a\,\mathrm{sgn}(x) - x\|^2 = \frac{\|x\|_1^2}{d} - 2\frac{\|x\|_1^2}{d} + \|x\|^2 = \|x\|^2 - \frac{\|x\|_1^2}{d} \leq \|x\|^2 - \frac{\|x\|^2}{d} = s\|x\|^2
$$

for $s = \left(1 - \frac{1}{d}\right)$. The first equality is obtained by noting that $\|\mathrm{sgn}(x)\|^2 = d$ and $\langle \mathrm{sgn}(x), x \rangle = \|x\|_1$. The inequality is due to $\|x\| \leq \|x\|_1$. Note that $a\,\mathrm{sgn}(x)$ is the Scaled-sign compressor, which is a $\frac{1}{d}$-approximate compressor since $0 < \frac{1}{d} \leq 1$.

For the first term in (2) we have

$$
\begin{aligned}
\mathbb{E}[a^2\|\mathrm{sgn}(y) \odot (2b - 1) - \mathrm{sgn}(x)\|^2] &= a^2 \sum_{j=1}^{d} \mathbb{E}[(\mathrm{sgn}(y[j])(2b[j] - 1) - \mathrm{sgn}(x[j]))^2] \tag{3} \\
&= a^2 4dq\alpha \\
&\leq 4q\alpha\|x\|^2,
\end{aligned}
$$

where $q$ is as defined in Algorithm 2. Recall that $q$ is the fraction of elements in $x$ such that $\mathrm{sgn}(x[j]) = \mathrm{sgn}(y[j])$. The last step is obtained by noting that $a^2 d = \frac{\|x\|_1^2}{d} \leq \|x\|^2$. The second equality is obtained with the following reasoning. The $d$ elements in summation can be split into three sets. In the first set we have $d(1 - q)$ elements satisfying $\mathrm{sgn}(x[j]) \neq \mathrm{sgn}(y[j])$, which will cause $b[j] = 0$ and $\mathrm{sgn}(y[j])(2b[j] - 1) - \mathrm{sgn}(x[j]) = 0$. The rest of the $dq$ elements satisfy $\mathrm{sgn}(x[j]) = \mathrm{sgn}(y[j])$ and can be further divided into two sets. Out of these $dq$ elements, in expectation an $1 - \alpha$ fraction have $b = 1$, again making $\mathrm{sgn}(y[j])(2b[j] - 1) - \mathrm{sgn}(x[j]) = 0$. Such elements make up the second set. The third set consists of the rest of the elements. Specifically, out of the $dq$, we have in expectation an $\alpha$ fraction with $b = 0$. For the elements in this third set we have $\mathrm{sgn}(y[j])(2b[j] - 1) \neq \mathrm{sgn}(x[j])$. This gives us $(\mathrm{sgn}(y[j])(2b[j] - 1) - \mathrm{sgn}(x[j]))^2 = 4$ due to the sign mismatch.

For the last term in (2) we have

$$\mathbb{E}[a\|\mathrm{sgn}(y) \odot (2b - 1) - \mathrm{sgn}(x)\|] = a\mathbb{E}\left[\left(\sum_{j=1}^{d}(\mathrm{sgn}(y[j])(2b[j] - 1) - \mathrm{sgn}(x[j]))^2\right)^{1/2}\right]$$

$$\leq \left(a^2\sum_{j=1}^{d}\mathbb{E}[(\mathrm{sgn}(y[j])(2b[j] - 1) - \mathrm{sgn}(x[j]))^2]\right)^{1/2}$$

$$\leq \sqrt{4q\alpha}\|x\|.$$

The first inequality is obtained by applying Jensen's inequality to the concave square root function, and the second inequality follows from (3).

Now we substitute the bounds for all terms in (2) to get

$$\mathbb{E}[\|C_\alpha^y(x) - x\|^2] \leq (4q\alpha + s + 2\sqrt{4q\alpha}\sqrt{s})\|x\|^2 \leq (\sqrt{4\alpha} + \sqrt{s})^2\|x\|^2.$$

In the last step we remove the dependency on $q$ as it is a function of $x$. The inequality is obtained by noting that $q \leq 1$ for all $x$ and $y$. Finally, letting $(\sqrt{4\alpha} + \sqrt{s})^2 = 1 - \delta$ we find the conditions for $\alpha$ so that $0 < \delta \leq 1$. Note that $\delta$ trivially satisfies $\delta \leq \frac{1}{d} < 1$ for all $\alpha \geq 0$. The lower bound of $\delta$ is satisfied if $0 < 1 - (\sqrt{4\alpha} + \sqrt{s})^2$ which yields the bound $\alpha < \frac{1}{4}(1 - \sqrt{s})^2$. This concludes the proof.

$\square$

## A.2 Proof of Theorem 2

We start with the proof of Theorem 1 by Zheng et al. (2019) provided in their Appendix C. In the following analysis we adhere to the same notation as in Theorem 1 to make the comparison easier. The following equivalencies hold in their and our notation: $t = k$, $\eta_t = \eta_k$, $x_t = w_k$, $M = n$, $\tilde{e}_t = \frac{1}{\eta_{k-1}}e_k$, and $e_{t,i} = \frac{1}{\eta_{k-1}}e_k^i$.

*Proof.* At the end of the first half of the proof on page 14 (Zheng et al., 2019), for $\rho > 0$, authors obtain

$$\mathbb{E}[F(\tilde{x}_{t+1})] \leq F(\tilde{x}_t) - \eta_t\left(1 - \frac{L\eta_t + \rho}{2}\right)\|\nabla F(x_t)\|^2 + \frac{L\eta_t^2\sigma^2}{2M} + \frac{\eta_t\eta_{t-1}^2 L^2}{2\rho}\left\|\tilde{e}_t + \frac{1}{M}\sum_{i=1}^{M}e_{t,i}\right\|.$$

Let us define $\lambda = 1/\rho > 0$ and $c = 1 - \frac{1}{2\lambda}$. By taking total expectation and then applying Lemma 6 by Zheng et al. (2019) with $\mu = 0$ we get

$$\mathbb{E}[F(\tilde{x}_{t+1})] \leq \mathbb{E}[F(\tilde{x}_t)] - \eta_t\left(c - \frac{L\eta_t}{2}\right)\mathbb{E}\left[\|\nabla F(x_t)\|^2\right] + \frac{L\eta_t^2\sigma^2}{2M}$$

$$+ \frac{\eta_t\eta_{t-1}^2 L^2\lambda}{2}\frac{8(1 - \delta)G^2}{\delta^2}\left(1 + \frac{1}{\delta^2}\right).$$

Taking telescoping summation over $T$ iterations and rearranging the terms give

$$\mathbb{E}\left[\min_{t=0,\ldots,T-1}\|\nabla F(x_t)\|^2\right] \leq \frac{F(x_0) - F_* + \frac{L\sigma^2}{2M}\sum_{t=0}^{T-1}\eta_t^2 + \frac{4\lambda L^2(1-\delta)G^2}{\delta^2}\left(1 + \frac{1}{\delta^2}\right)\sum_{t=0}^{T-1}\eta_t\eta_{t-1}^2}{\sum_{t=0}^{T-1}\eta_t\left(c - \frac{L\eta_t}{2}\right)},$$

where we have asserted that $\mathbb{E}[F(\tilde{x}_0)] = F(x_0)$ and that $\mathbb{E}[F(\tilde{x}_T)] \geq F_*$. Next we pick the step size schedule $\eta_t = \frac{c}{L\sqrt{T}}$ and rearrange terms to get

$$\mathbb{E}\left[\min_{t=0,\ldots,T-1}\|\nabla F(x_t)\|^2\right] \leq \frac{\frac{2L}{c^2}(F(x_0) - F_*) + \frac{\sigma^2}{M}}{2\sqrt{T} - 1} + \frac{\frac{8c\lambda(1-\delta)G^2}{\delta^2}\left(1 + \frac{1}{\delta^2}\right)}{2T - \sqrt{T}}. \quad (4)$$

In comparison to (4), the analogous bound for SGD is

$$\mathbb{E}\left[\min_{t=0,\ldots,T-1}\|\nabla F(x_t)\|^2\right] \leq \frac{2L(F(x_0)-F_*)+\frac{\sigma^2}{M}}{2\sqrt{T}-1}. \tag{5}$$

For large $T$, the second term in (4) vanishes at a faster rate than the first term. However, we cannot make a direct comparison between (5) and the first term in (4) when $c^2 \neq 1$. Setting an arbitrarily large value for $\lambda$ gives $c^2 \approx 1$, but we pay a penalty with a large $c\lambda$ in second term in (4). The solution is to make $c$ asymptotically converge to 1 by carefully choosing a $\lambda$ that grows with $T$. This way $c \to 1$ and the first term in (4) becomes directly comparable to (5). We also want the $c\lambda$ in the second term of (4) small enough so that it vanishes faster than the first term in (4).

Let $\lambda = T^s$ for some $s > 0$. We solve for $s$ by expanding $c\lambda$ and $\frac{1}{c^2}$ in (4). We have $c\lambda = \lambda - \frac{1}{2} = T^s - \frac{1}{2}$. We take the following approach to expand $\frac{1}{c^2}$. For $z \in \mathbb{R}$ and positive integer $r$, from the negative binomial series we have

$$(1-z)^{-r} = \sum_{p=0}^{\infty} \binom{r+p-1}{p} z^p.$$

For $r = 2$ and $|z| < 1$ we can write

$$(1-z)^{-2} = 1 + 2z + 3z^2 + 4z^3 + \cdots < 1 + 2z + \mathcal{O}(z^2).$$

Since $\frac{1}{2\lambda} = \frac{1}{2T^s} < 1$, we can write

$$\frac{1}{c^2} = \left(1 - \frac{1}{2\lambda}\right)^{-2} < 1 + \lambda^{-1} + \mathcal{O}\left(\lambda^{-2}\right) = 1 + T^{-s} + \mathcal{O}\left(T^{-2s}\right).$$

Substituting the result back in (4) and rearranging terms gives

$$
\begin{aligned}
\mathbb{E}\left[\min_{k=0,\ldots,T-1}\|\nabla F(x_t)\|^2\right] &< \frac{2L(F(w_0)-F_*)+\frac{\sigma^2}{M}}{2\sqrt{T}-1} \\
&+ \left(T^{-s} + \mathcal{O}\left(T^{-2s}\right)\right)\frac{2L(F(w_0)-F_*)}{2\sqrt{T}-1} \\
&+ \left(T^s - \frac{1}{2}\right)\frac{1}{2T-\sqrt{T}}\frac{8(1-\delta)G^2}{\delta^2}\left(1+\frac{1}{\delta^2}\right) \\
&< \frac{2L(F(w_0)-F_*)+\frac{\sigma^2}{M}}{2\sqrt{T}-1} \\
&+ \mathcal{O}\left(T^{-s-\frac{1}{2}}\right) + \mathcal{O}\left(T^{-2s-\frac{1}{2}}\right) + \mathcal{O}\left(T^{s-1}\right) + \mathcal{O}\left(T^{-1}\right).
\end{aligned} \tag{6}
$$

We would like to pick an $s$ that makes all terms in the last row go to zero in the same asymptotic rate. The third term requires $s < 1$. For $0 < s < 1$ the first and the third terms are the slowest to go to zero, therefore, they determine $s$. Setting $-s - \frac{1}{2} = s - 1$ gives $s = \frac{1}{4}$. One can substitute $s = \frac{1}{4}$ in (6) to obtain

$$
\begin{aligned}
\mathbb{E}\left[\min_{k=0,\ldots,T-1}\|\nabla F(x_t)\|^2\right] &< \frac{2L(F(w_0)-F_*)+\frac{\sigma^2}{M}}{2\sqrt{T}-1} \\
&+ \frac{2L(F(w_0)-F_*)+\frac{8(1-\delta)G^2}{\delta^2}\left(1+\frac{1}{\delta^2}\right)}{2T^{3/4}-T^{1/4}} + \mathcal{O}\left(\frac{1}{T}\right),
\end{aligned}
$$

where we have incorporated two $\mathcal{O}\left(\frac{1}{T}\right)$ into one. Changing the notation to our original yields the result in (1), concluding the proof.

$\square$

## A.3 PERFORMANCE OF LZMA COMPRESSION

Figure 3 compares the compression ratio of LZMA algorithm with the theoretical lower bound. We generate a binary sequence of length $10^7$ by sampling from the Bernoulli distribution with probability $p$, compress the sequence using LZMA, and plot the compression ratio versus $p$. The entropy $H(p)$ is a lower bound for the compression rate.

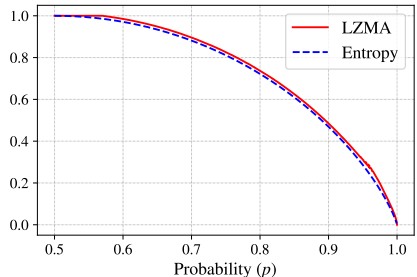

Figure 3: Compression performance of the LZMA algorithm on a random binary sequence.

## A.4 IMPACT OF BATCH SIZE ON TEMPORAL CORRELATION

The stochastic gradient workers compute is noisy. This is because workers use only a small minibatch out of a large dataset. We expect the gradient to be less noisy when the size of the minibatch is larger, and we expect less noisy gradients to be more (positively) correlated across time. First, note that as per Algorithm 1, $\hat{g}_k$ is the sum of the stochastic gradient and the compression error in the last iteration. Second, as noted in Algorithm 2 $q$ is the fraction of entries such that $\text{sgn}(\hat{g}_k[j]) = \text{sgn}(\bar{g}_k[j])$. Although $\hat{g}_k$ and $\bar{g}_k$ are not stochastic gradients themselves, we expect their correlation (measured with $q$) to be impacted by the minibatch size. As $q$ is central to our proposed compression scheme, we explore numerically how $q$ changes with the minibatch size. Recall that we first observed $q$ in the central sub-figure in Figure 2. To this end we rerun the ImageNet32-SignXOR experiments described in Section 5.1 for 4 different minibatch sizes, and for two values of $\alpha$. Figure 4 summarizes the new results plotted versus the step. The first and second rows correspond to experiments with $\alpha = 0$ and $\alpha = 0.7$ respectively. Training is started with a learning rate of $10^{-4}$, and the learning rate is multiplied iteratively by $0.1$ at certain epochs when the loss starts to plateau. These epochs are multiples of 10, 15, 30 and 35 for the experiments with batch sizes 16, 32, 64, 128 respectively.

We observe that for large minibatch sizes the loss decreases faster. This is because the gradients are less noisy for large minibatches. We also observe that a larger minibatch result in a larger $q$. The reason for this observation is that, as stochastic gradient gets less noisy, the stochastic gradients within $\hat{g}_k$ and $\bar{g}_k$ get more (positively) correlated (recall that $\hat{g}_k$ is the sum of the stochastic gradient and the compression error). This results in an increment of $q$, the fraction of components in $\hat{g}_k$ and $\bar{g}_k$ that share the sign. We also observe in Figure 4 that this increment is minimal. For $\alpha = 0.7$ we only observe a maximum difference of around $0.005$ between the two $q$ plots corresponding to batch sizes 16 and 128. In summary, $q$ is less responsive to the changes of the batch size.

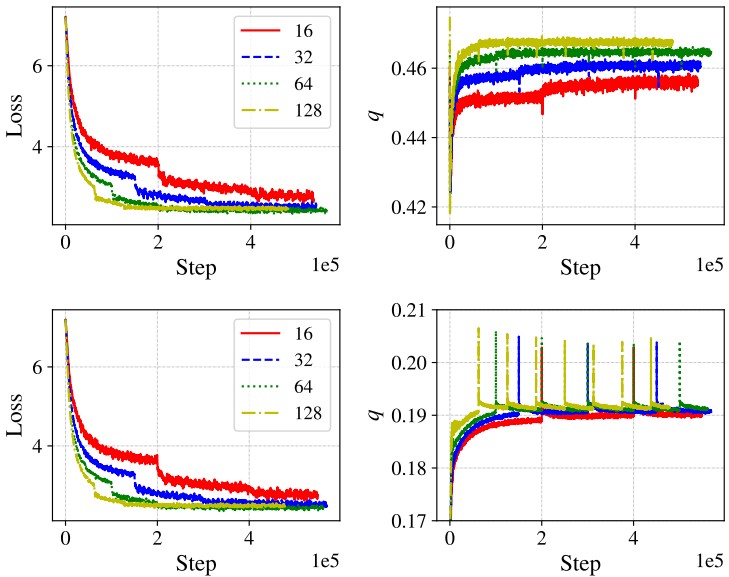

Figure 4: Assessing the impact of batch size on $q$ in SignXOR. The results from four different minibatch sizes are presented as indicated in the legends. The first and second rows correspond to experiments with $\alpha = 0$ and $\alpha = 0.7$ respectively.

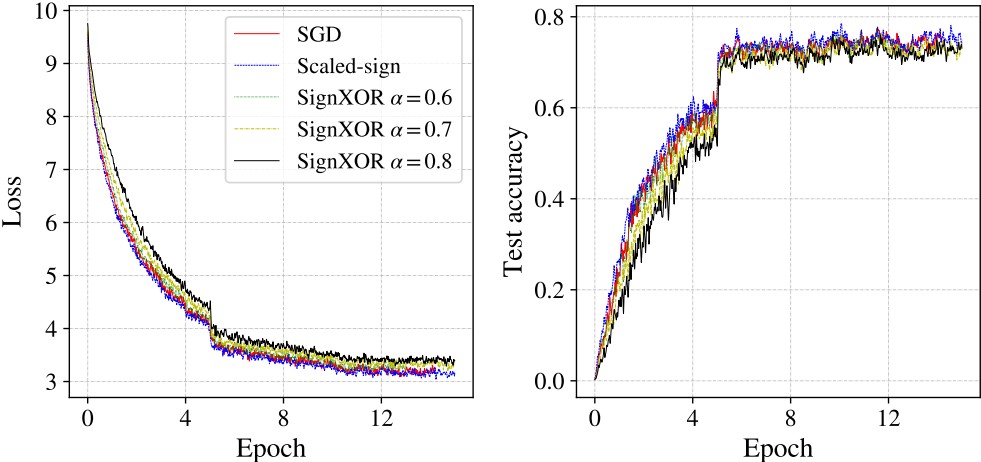

Figure 5: This figure is same as the first two sub-figures in the third column in Figure 1 except that the $x$-axis runs for a larger number of epochs.

