# OpenReview forum: "Compressing gradients in distributed SGD by exploiting their temporal correlation"
_ICLR.cc/2021/Conference — Reject_

### Official Review · AnonReviewer1 · 2020-10-20
**Official Blind Review #1**

**Rating:** 6
**Confidence:** 4

**Review:**

This paper proposed an extension of blockwise scaled sign compressor in Zheng et al. (2019). The proposed method exploits the temporal correlation between two consecutive gradients. The authors show that one can have a higher compression rate by inserting distortion to the compressed gradient. A tighten bound is provided such that the asymptotic rate (including constant) is exactly the same as the full-precision counterpart. The experiments show that the proposed compressor can achieve additional 40%-50% reduction on communication compared to the scaled sign. Overall, the reviewer thinks the idea is interesting. The reviewer has a few comments:

1. The proposed method considers randomly flipping the direction for elements that have the same sign as the averaged gradient in the last step. In this way, the sign is always correct for the elements that have opposite direction from the last gradient. I wonder will the results change if we consider flipping the sign of the elements that have opposite direction?

2. Since alpha has a very small upper bound, it is hard to see any theoretical improvement over scaled sign.

3. Theorem 4.2 does not show that one can achieve a linear speedup, i.e., O(1/\sqrt{nT}) rate.

4. For the distributed training with high speed network, the extra overhead incurred by compression is not trivial and cannot be overlooked. As there is no results against CPU wall clock time, it is not clear if the proposed method is really faster than the scaled sign in terms of elapsed time.

5. Can you show the final test accuracies on ImageNet achieved by each algorithm? It seems that scaled sign has slightly higher accuracy.

---

> ### Author Response · Authors · 2020-11-17
> **We thank the reviewer for the comments and the questions.**
>
> In the following we address all comments/questions in the same order.
>
> 1. There are two reasons for our particular choice of inverting the signs.  We will first summarize the general idea, and then give some specifics in relation to the numerical results provided. The high-level reason for choosing the quantizer we do is that (cf., middle plot Figure 2) even without quantization $q\approx0.45 < 0.5$.  Since the binary entropy function is symmetric around $q=0.5$ and is concave, for a fixed amount of distortion we get more rate savings by designing a quantizer that drives $q$ towards 0 rather than toward 1.  If we drive $q$ towards 1 we would need to get over the binary entropy “hump” at 0.5 and to 0.55 before we would start to incur rate savings. By that point we would already have incurred non-negligible distortion.  For this reason, we design the quantizer to push $q$ towards zero.
>
>     (i) In the third sub-figure in Figure 2 we plot $p$, the fraction of $1$'s in vector $b$ for ImageNet32 experiments. Recall that $\alpha=0$ case recovers the vanilla scaled-sign compression. As per the third sub-figure, $p$ for $\alpha=0$ stays at around $0.45$. We observed that $p$ stays close to but slightly lower than $0.5$ for CIFAR100 experiments as well. We gain compression savings if the binary entropy $H(p)$ is closer to $0$. The binary entropy is symmetric around $0.5$. Therefore, we have to choose between two strategies: (A) push $p$ towards $0$, or, (B) push $p$ towards $1$ by introducing errors to vector $b$.
>
>     The former (A) corresponds forcing some of the components with same sign to be different, and the latter (B) corresponds to forcing some of the components with different signs to be the same. Since for $\alpha=0$ (no errors) we already have $p$ less than $0.5$ (say $0.45$) we make the choice to push $p$ towards $0$ by applying (A). Let us assume we were to use (B) instead. This means that we will have to introduce errors just to get $p>0.5$. For example, if we were to end up with $p=0.55$, we will have obtained no gain in terms of compression (since $H(0.45)=H(0.55)$), although we would already have introduced some errors to $b$ vector.
>
>     (ii) We also note that, to confirm our design idea, during our research we also ran experiments that pushed q toward 1.  While we did not include those results in the final paper, those systems worked *less* well due to our reasoning in (i).
>
>
> 2. We agree with the reviewer that the upper bound is small. This concern was also raised by the third reviewer (AnonReviewer3) and we responded at length to that reviewer. We ask this reviewer to refer to that response (please see under the *Addressing the comment about Theorem 1* heading) which we are not reproducing here to keep our response more compact.
>
> 3. This is due to our choice of the step size. The choice of step size by Zheng et al. (2019) yields two bounds for dist-EF-SGD and SGD as are presented in Corollary 1. We observe that although they achieve a linear speedup, both first and the second terms in the dist-EF-SGD bound are larger by constant factors compared to those of the SGD bound (first two terms in dist-EF-SGD have 4 and 1, whereas first two terms in SGD have $\frac{8}{3}$ and $\frac{2}{3}$). In comparison, our choice of step size gives us identical first terms for the proposed method and SGD (our Theorem 2). Not achieving a linear speedup is the price we pay for getting the first terms to match.
>
> 4. We agree that the compression overhead may be non-trivial. However, our primary goal is to showcase the reduction of the communication payload. All the steps in in Algorithm 2 can be efficiently implemented using vectorized operations except lossless compression (and decompression) of vector $b$. The lossless compression overhead heavily depends on the algorithm and the implementation of the algorithm. We employ the Python implementation of the LZMA algorithm in our experiments. Since our setup does not use the most computationally efficient implementation of compression, we do not think it is fair to compare the compression overheads involved and therefore did not include such a comparison.
>
> 5. In our original experiments we ran SignXOR only for a slightly greater than 10 epochs due to resource limitations. As per the reviewer’s request, we have rerun SignXOR $\alpha=0.6$ for a longer time and included an updated figure in Appendix (Figure 5 on last page in the updated manuscript). Currently it shows SignXOR $\alpha=0.6$ only up to the 12th epoch (still running past the 12th epoch as of the time of this response). We hope to update the final test accuracies prior to the end of the rebuttal period.

---

> > ### Author Response · Authors · 2020-11-23
> > **This response is a follow-up to (5.) in our earlier response**
> >
> > 5. UPDATE: We have rerun all ImageNet experiments for a longer time and included an updated figure in Appendix (Figure 5 on last page in the updated manuscript). It shows the performance of all algorithms up to the 15th epoch. We notice that Scaled-Sign achieves a slightly higher accuracy than SignXOR. However, we also note that one can achieve Scaled-Sign accuracy with SignXOR by decreasing $\alpha$ towards the end of the training (i.e., reduced distortion). This means that we start SignXOR with a higher $\alpha$ (e.g. $\alpha=0.6$), and decrease $\alpha$ for a few epochs towards the end of training (e.g. $\alpha=0.1$). A smaller $\alpha$ behaves more like Scaled-Sign due to reduced distortion. This method will incurr for the last few epochs a communication cost similar to that of Scaled-Sign. However, the increment of communication cost for that few epochs will be low compared to the total communication budget for 15 epochs.

---

### Official Review · AnonReviewer3 · 2020-10-27
**Interesting approach to combine sign-based gradient compression with lossless encoding. Not convinced about the 'temporal correlation'.**

**Rating:** 4
**Confidence:** 4

**Review:**

This paper proposes a gradient compression approach to remove the communication bottleneck in distributed stochastic gradient descent. I think the key attributes of their algorithm are as follows:
- It uses error feedback (Stich et al., 2018; Karimireddy et al., 2019) and operates in a parameter server model based on (Zheng et al. 2019)
+ The compressor sends the sign of each gradient coordinate and a scale factor per 'block' of coordinates (like Zheng et al. 2019)
+ The messages are compressed with lossless entropy encoding.
+ Specifically, they send and encode the difference between the current sign vector and the previously transmitted one. This is meant to lower the entropy of the vectors (make the distribution of -1's and 1's less even)
+ Because the distribution of -1's and 1's is still roughly 50/50 after delta-coding, the authors introduce (lossy) noise in the compressor. They randomly flip some instances of 'same sign as before' to 'different sign than before'. This reduces the entropy of the 'difference vector' so it can be compressed more.

I believe the main contributions of the paper are:
- The introduction and evaluation of lossless compression on top of sign-based gradient compression
- A theoretical improvement of the constants in the rates from (Zheng et al. 2019)
- A proof that SignSGD with delta-coding and a bias towards changing signs can still be a $\delta$-compressor, as long as the bias is extremely small (not covering the experiments presented in the paper)

I find the ideas presented in this paper interesting and novel and the experiments well executed. The writing is of good quality, and I find it easy to follow. I do, however, have two concerns:
- The method is said to exploit temporal correlation in the gradients by using delta coding. To me, this seems misleading. The authors show that, without the introduced bias, there is not much gain from delta coding ("In our experiments presented in Section 5.2 we observe that when α = 0, p remains close to but slightly lower than 0.5. This implies low correlation between [..]. Our solution is to make $\alpha > 0$."). With the proposed solution, the method exploits patterns created artificially by lossy compression, rather than temporal correlation in the gradients. Given that p < 0.5, this is actually anti-correlation rather than correlation. The signs are more likely to flip due to previously introduced errors.
- If the delta-coding scheme indeed fails to leverage temporal correlation in the gradients without artificially introducing extra errors, the proposed scheme doesn't really do what it seems to be designed for. This makes it less elegant/complicated. This lack of simplicity could be compensated by convincing experimental results, but it seems that many gradient compression schemes achieve similar results to the proposed scheme at similar compression rates (see Xu et al. https://repository.kaust.edu.sa/bitstream/handle/10754/662495/gradient-compression-survey.pdf) for an overview). I am not convinced by the benefits of the proposed scheme over others.

---

> ### Author Response · Authors · 2020-11-17
> **We address the comments on temporal correlation and Theorem 1**
>
>
> ***Addressing the temporal correlation-related comments***
>
> We thank the reviewer for the comments. The issue raised by the reviewer regarding temporal correlation is very important one that was also shared by the first reviewer (AnonReviewer4) and we responded at length to that reviewer. We ask this reviewer to refer to that response (please see *Part 1/2: Addressing concerns regarding the source of rate savings*) which we are not reproducing here to keep the response more compact. A summary of that response is as follows.
>
> We demonstrate using our experimental results that the total compression gain is due to (i) lossy compression and (ii) temporal correlation. We argue that the temporal correlation is not because of the lossyness (or distortion) introduced in the current step, rather due to the introduction of distortion in previous steps. This yields the temporal correlation with the next iteration that we then leverage.  In addition to this temporal correlation we add distortion to the current step which gives us more compression.  As discussed in the response to AnonReviewer4, the issues of lossy compression and leveraging temporal correlation are intertwined in our system.
>
>
>
> ***Addressing the comment about Theorem 1***
>
> We would also like to respond to the reviewer’s comment about the upper bound for bias ($\alpha$). We agree with the reviewer that the upper bound is small. The reason for this is the extremely small $\delta$ that corresponds to the scaled-sign compression. We attribute this to the mismatch one often observes between theory and experiments where the assumptions that need to be made to make the theory work are often more conservative than those that are often observed to work in practice. Next we reason why this is the case in our proposed scheme.
>
> Recall that scaled-sign is a $\frac{1}{d}$-compressor where $d$ is the dimension of the input vector. Let us consider the EF-SignSGD algorithm by Karimireddy et al. (2019). The convergence bound of EF-SignSGD is given in their Remark 4. In summary, the difference between the bounds of EF-SignSGD and vanilla SGD is that the former has an additional term with $1/\delta^2$. Since $d$ is in the millions for models of practical interest, we expect the $1/\delta^2$ term to be extremely large and to have a large impact on the convergence. However, Karimireddy et al. (2019) show in their experiments that EF-SignSGD performs very close to vanilla SGD from the beginning of the training. In summary, although one expects the convergence to slow down due to the small $\delta$ of scaled-sign, it is not quite the case in practice.
>
> This is similar to what we observe in our scheme. The proof of our Theorem 1 relies on the fact that scaled-sign is a $\frac{1}{d}$-compressor. This constrains the wiggle room for the proposed system to be a $\delta$-compressor (since we are building on scaled-sign), which in turn limits the bound on $\alpha$. However, our experiments demonstrate that contrary to the theoretical upper bound, one can set $\alpha$ to be larger and the algorithm can still converge.

---

> > ### Comment · AnonReviewer3 · 2020-11-24
> >
> > Thank you for addressing my comments.
> >
> > - I agree with the authors that the discrepancy between theory and practice is common. It is unfortunate, but I don't consider this a significant problem.
> > - Regarding 'temporal correlation': as the authors point out, the (negative) correlation stems from distortions introduced by the algorithm in the previous gradient updates. With this, I maintain that message of the paper (including the title) that the algorithm exploits temporal correlation in the gradients is misleading. The idea of delta coding is less interesting if its gain comes from manually introduced distortions.

---

### Official Review · AnonReviewer2 · 2020-10-29
**The authors proposed a bidirectional gradient compression scheme called SignXOR for distributed training.**

**Rating:** 2
**Confidence:** 5

**Review:**

1.	The main problem I have with this paper is that this paper idolizes the paper [distributed EF-SGD by Zheng et al. NeuRIPS 2019]. However, the main result, that is, Theorem 1 in dist-EF-SGD is mathematically problematic or simply wrong. The proof of Theorem 1 as given in [distributed EF-SGD by Zheng et al. NeuRIPS 2019] does not hold good when the learning rate sequence $\eta_t>0$ is decreasing. Therefore, the authors’ claim in the Abstract and several parts of the present paper “We strengthen their analysis to show that the rate of convergence of two-way compression with error- feedback asymptotically is the same as that of SGD” eventually invalid. I suggest the authors please read the distributed-EF-SGD paper carefully, understand it better, write the proofs on their own before making these types of strong claims. Please study [1] and work on your proofs.
2.	Page 2: “However, Karimireddy et al. (2019) theoretically show that SGD with compression does not converge in general.” This is a very strong statement which I do not agree with. Karimireddy et al. (2019) showed how error feedback can fix the convergence issue of sign-based quantization as in signSGD. Moreover, they showed how any compressor (biased/unbiased) can be converted to a $\delta$-compressor. But that does not mean the authors’ statement in this manuscript is correct. As authors claimed “error feedback-based algorithms circumvent the convergence issues for SGD with compression” is not right. Error feedback is known to work well for sparsification, where a subset of gradient components are sent or for extreme quantization (such as sign-based compressions). However, regular random dithering-based quantization techniques such as QSGD, natural compression, etc. converge just fine without error feedback. Actually, error feedback may degrade their performance. I would like to request the authors to first understand these works in detail before writing these types of strong statements on their paper.
3.	Another vague statement is: “Therefore, these two classes can be respectively thought of as approaches that reduce the quantity versus the quality of the gradient.” Based on what you claimed this?
4.	“Sign-based compression schemes such as Scaled-sign, SignSGD and Signum (Bernstein et al., 2018)…” SIGNUM is not a sign-based compressor. It is the momentum version of signSGD, nothing novel.
5.	You may want to talk about the most relevant and recent work on compression known as SketchML [J.  Jiang,  F.  Fu,  T.  Yang,  and  B.  Cui,  “SketchML:  Accelerating Distributed Machine Learning with Data Sketches,” SIGMOD, 2018] while talking about delta encoding first paragraph in Section 3. This recent paper on gradient compression uses delta encoding.
6.	I failed to understand the benefit of Generalized dist-EF-SGD algorithm in Section 3? What are the main differences telling me?
7.	“In our case, while the length of the parameter vector d is well over a million for models of practical interest, the entries of b are not necessarily i.i.d..” Can you make an assumption of the independence of the gradient components? If you make that assumption you may elevate the issue. Also, the assumption is not a strong assumption and generally made for stochastic gradients. Please See [Huffman Coding Based Techniques for Fast Distributed Deep Learning, Gajjala et al., CoNext DistML workshop, 2020]
8.	Uniform upper bound of the stochastic gradients g_i is an obsolete concept. The authors may argue that "The classical theoretical analysis of SGD assumes that the stochastic gradients are uniformly bounded". But one can even strongly argue that this bound is actually $\infty$. Moreover, an even a stronger argument can be made that the above assumption is in contrast with strong convexity. Please see ["SGD and Hogwild! Convergence Without the Bounded Gradients Assumption" by Nguyen et al.] as one of the instances. Please understand there are relaxed assumptions such as Strong growth condition on a stochastic gradient as in Assumption 4 of [2].
9.	You said: “Since the communications component in Horovod is designed for a master-less setup, we simulate a master-worker environment in our implementation.” But I was wondering why do you need this? In any case, if you use all-reduce collective for aggregation even for P2P architecture the aggregation will be similar. Please correct me if I am wrong.
10.	While ImageNet accuracy is similar to why test accuracies of the models on CIFAR-100 and ImageNet-32 are below 60%?
11.	In terms of experimental results, instead of Figure 1, the authors may use relative data-volume vs. test accuracy. Especially, when the accuracy figures are really cluttered. To do proper experiments by using compression techniques, the authors can check a very elaborative work and codebase by [Hang Xu et. al, Compressed Communication for Distributed Deep Learning: Survey and Quantitative Evaluation.] In that case, I would encourage the authors to plot relative data-volume vs. test accuracy similar to Figures 6 and 7 therein. I am sorry but in the present papers, the experiments and their presentations are substandard.
12.	 Why did not you compare with sign-sgd algorithm? Also, sign-based algorithms are notorious in their convergence. SignSGD uses a special stepsize. So I was wondering what is your step-size schedule? You never mentioned this in your paper. You may did it in the Appendix and I did not check the Appendix. So, please indicate if you did.


Minor Comments:

1.	These two sentences in my understanding are claiming the same thing?
“We strengthen their analysis to show that the rate of convergence of two-way compression with error feedback asymptotically is the same as that of SGD. As a corollary, we prove that two-way SignXOR compression with error-feedback achieves the same asymptotic rate of convergence as SGD.”
2.	“Novel approaches such as federated learning…” Why is it novel? Unnecessary hyperbole is not part of technical writing.
3.	“In this section, we prove that the combination of Algorithm 1 and Algorithm 2 converges, and the convergence rate is asymptotically the same as that of SGD.” Why Algorithm 2 when Algorithm 1 implicitly implies the inclusion of Algorithm 2?
4.	Please correct the typos and please write as it is done in a technical paper, not in a sci-fi novel.

[1] Communication-Efficient Distributed SGD with Error-Feedback, Revisited, Tran Thi Phuong, Le Trieu Phong, 2020.
[2] Dutta et al. AAAI 2020, On the Discrepancy between the Theoretical Analysis and Practical Implementations of Compressed Communication for Distributed Deep Learning

---

> ### Author Response · Authors · 2020-11-17
> **Part 2/2: Please read Part 1/2 prior to reading this**
>
>
> 9. In Algorithm 1 we assume a master-worker setup with two-way compression. To be consistent with Algorithm 1, we simulate a master-worker setup in experiments even though the underlying Horovod implementation is based on allreduce.
>
>     If we use the allreduce aggregation as the reviewer suggests, workers will compress before calling allreduce. Allreduce computes the average of compressed gradients, which will be used to update the parameter vector. This is equivalent to one-way compression in a master-worker setup. However, we are considering two-way compression.
>
> 10. The reduction of test accuracy in ImageNet-32 is to be expected as it is a down-sampled version of ImageNet, therefore, comprising of less informative images. Chrabaszcz et al., 2017 provide a baseline for ImageNet-32. As can be observed in their Figure 2, the final test accuracy varies between 70% and 60% for different learning rates. In comparison, we achieve a 60% accuracy with SGD while they achieve a higher accuracy as they use SGD with momentum.
>
>     For CIFAR-100 we use Zheng et. al. 2019 as the baseline. The difference between their CIFAR-100 results and ours is due to two reasons: (i) we use resnet-18 and they use resnet-20, and (ii) we use a weight regularizer of $10^{-3}$ whereas they use $5\times10^{-4}$. In any case, we note that our goal is to compare the relative performances of the algorithms under *same experimental settings*. Therefore, we do not perform an exhaustive search over hyperparameters that achieve state-of-the-art error rates.
>
> 11. We do not see how the reviewer’s suggestion of plotting data-volume vs. test accuracy provides much more information than what can already be extracted from the existing figures. The test accuracy and loss plots are clear enough to see the general trend. The $B_X/B_S$ plots show versus epoch the relative data usage of the proposed method (relative to Scaled-Sign). We chose the presentation style in Figure 1 because it provides information about loss/test accuracy with respect to both relative data usage as well as the epoch. If, as in some of the literature, a much larger set of algorithms were being considered then we agree with the reviewer that scatter plots of data-volume vs. final test accuracy can quite nicely summarize where algorithms stand in relation to each other. To make the important points that we want to make in the current submission we believe the current plots serve our purposes well.
>
>     As one important example of how the plots serve our purposes, we direct the reviewer to our response to the AnonReviewer4 (please see *Part 1/2* therein). In response that that reviewer’s question we used our Figure 1 and particularly our Figure 2 to illustrate the combined effects of exploiting (i) the lossy compression approach we designed and (ii) the resulting temporal correlation induced to reduce the rate of communication. This is a key idea that we wanted to use the experiments to illustrate. We feel the experiments we provided accomplish that – though of course we thank the reviewers for asking the questions that helped us better to explain the ideas we wanted the experimental results to illustrate. We also note that our experiments and their presentation is similar to a number of other results in the field, e.g., those by Bernstein et al., 2018 (Figure 3), Karimireddy et al., 2019 (Figure 4), and Zheng et. al. 2019 (Figure 2,3).
>
> 12. Please refer to the first paragraph in Section 5.1 in the main text were have described the learning rate schedules for all experiments.
>
>     We chose not to compare with SignSGD as (1) the communication usage (per iteration) of SignSGD is same as that of its error-feedback counterpart (EF-SIGNSGD by Karimireddy et al., 2019), and (2) EF-SIGNSGD performs better than SignSGD as demonstrated by Karimireddy et al., 2019. We believe it suffices to compare with the better algorithm out of the two. This decision also kept our figures and discussion less cluttered.
>
>
>
> ***Minor comments***
>
> 1. The first sentence is about the convergence rate of error-feedback with *any* $\delta$-compressor.
>
>     The second sentence indicates that SignXOR with error-feedback converges as it is a $\delta$-compressor.
>
> 2. We are sorry the reviewer’s taste was not always well served by our writing style. We appreciate that there are variations in aesthetics and also appreciate that some of the other reviewers commented favorably on the style of the paper.
>
> 3. We agree with the reviewer that phrasing in this sentence may have been confusing. Our message is that *SignXOR compression (Algorithm 2) with generalized-EF-SGD (Algorithm 1) converges*. We have refactored the paragraph in the latest manuscript. The updated text is indicated in blue color.
>
> 4. We will correct all the typos indicated by all reviewers (thank you very much!) and will of course do another round of close reading if the paper is accepted for final publication.

---

> ### Author Response · Authors · 2020-11-17
> **Part 1/2: We thank the reviewer for the comments and the questions.**
>
> We thank the reviewer for the comments and questions. In the following we address all of them in the same order.
>
> 1. We believe a better phrasing would be that this paper "builds on" the work by Zheng et al. NeuRIPS (2019). We note that our Theorem 2 relies on a *constant* $\eta_k$ that is not a function of $k$. Therefore, we are *not* considering a decreasing learning rate, and the essence of our Theorem 2 does not change. Recall that the core message of Theorem 2 is that two-way compression with error-feedback asymptotically obtains the same convergence rate as SGD.
>
> 2. We think reviewer may have misinterpreted the ideas we were trying to convey due to our phrasing.  We detail our thinking anew below.
>
>     (A) *”However, Karimireddy et al. (2019) theoretically show that SGD with compression does not converge in general.”*
> We mean by (A) that *SGD with arbitrary compression schemes may not converge*. Karimireddy et al. (2019) give one such example of non-convergence using sign as the compression scheme. We have rephrased the sentence in the manuscript to avoid confusion. The updated text is indicated in blue color.
>
>     (B) *"error feedback-based algorithms circumvent the convergence issues for SGD with compression"*
> Karimireddy et al. (2019) prove (in their Theorem II and Remark 4) that as long as a $\delta$-compressor is used, the application of error-feedback guarantees the convergence of the algorithm. The sentence (B) is simply a rewording of this fact. We note that we have not mentioned $\delta$-compressors in the quote (B) as (B) is from Section 1 and we do not introduce $\delta$-compressors until Section 2.
>
> 3. The two sentences that preceed the quoted sentence may provide the right context in which to understand what we were trying to say. Gradient sparsification reduces the number of non-zeros entries in the compressed gradient. This effectively reduces the **quantity** of information carrying gradient components. On the other hand, gradient quantization reduces the **precision** of individual gradient components.  This effectively reduces their quality. One can think of this in analogy to image compression. JPEG determines both which transform coefficients to store (the quantity) and at what level of resolution to store those coefficients (the quality). The sentence the reviewer quotes is an abstraction of this classical distinction.
>
>     If the paper is accepted for publication, to avoid any ambiguity (thank you for pointing that out) we will include the above reasoning in the manuscript as we get one extra page.
>
> 4. We observe that in this sentence we have nowhere claimed "novelty" of Signum. We are simply suggesting that compression offered in Signum is due to the application of the sign function. This can be verified by inspecting Algorithm 2 by Bernstein et al., (2018).
>
> 5. We thank the reviewer for the suggestion. We have updated the relevant paragraph in Section 3 to reflect this.
>
> 6. We have summarized these main differences in the **Generalized dist-EF-SGD** sub-section in Section 3 (page 3), after the sentence *"The three main differences between the generalized and original dist-EF-SGD versions are as follows"*.
>
>     We have not claimed any benefit of the generalized dist-EF-SGD algorithm. As we note in the second last sentence just prior to the **Generalized dist-EF-SGD** section, *"The generalization is to make dist-EF-SGD compatible with SignXOR"*.  Effecting such compatibility was the reason we introduced Generalized dist-EF-SGD.  We are sorry for any confusion on this point.
>
> 7. It's not quite clear to us why the reviewer suggests to "make an assumption of the independence of the gradient components". When we say *"the entries of b are not necessarily i.i.d."*, we only indicate that the Shannon limit for i.i.d. sources is not directly applicable in our case. The Shannon limit gives a lower bound on the compressibility of a sequence of an i.i.d. observations. Since the entries of $b$ are not necessarily independent, $b$ must be at least as compressible as an iid source. We do not assume the independence as we do not rely on it.
>
> 8. We thank the reviewer and acknowledge that an improved analysis on error-feedback could be an interesting direction of future work. Herein our main objective was to study the interaction of the benefits that can be derived from (i) lossy compression and (ii) exploiting temporal correlation (as introduced by the design of our lossy compressor). We observe that the bounded gradient assumption is made in many recent publications on error-feedback such as Stich et al., 2018 (Theorem 2.4), Karimireddy et al., 2019 (Assumption C) and Zheng et. al. 2019 (Assumption 3).

---

### Official Review · AnonReviewer4 · 2020-11-02
**Interesting work but there appears to be a basic problem**

**Rating:** 5
**Confidence:** 4

**Review:**

The authors present a new scheme for compressing gradients for use in distributed training. In addition to the previously proposed techniques of sending the sign of the gradient components along with the scale, and the use of error feedback (each sender tracks the error introduced by quantization, and adjusts future gradient updates using it), the authors also propose to exploit the temporal correlation of gradient values (i.e., over successive steps). They do so by computing the delta between two steps, and then use a hyperparameter $\alpha$ to keep only a fraction of the deltas and that is sent losslessly.

The idea is an interesting (even if a fairly simple one) and leads to a greater than 50% savings.

However, I am confused about a basic issue. From the authors’ own data (Figure 2 and text) and from other research on alignment of per-example gradients (e.g. https://arxiv.org/abs/1901.09491, https://arxiv.org/abs/2008.01217), for the bulk of training, actual temporal correlation between gradients is quite low. So how can delta compression help?

(As an aside, the correlation is likely to be quite affected by batch size as per the second reference above, so some exploration/data around that would also be useful in the context of this proposal.)

This leads me to believe that the compression benefit they are seeing comes from higher values of $\alpha$, i.e., by throwing away information. So a natural question is: What happens if you simply do lossy compression on gradient signs? You would have to go to 3 values (+1, 0 and -1), and there would likely be some natural sparsity, and $\alpha$ could be used to enhance that. That would appear to be an interesting baseline to see how much benefit comes from the temporal aspect v/s the lossiness induced by $\alpha$. These two appear to be currently confounded. (Related: how does $\alpha$ = 0 look in Figure 1? Pretty bad from a compression point of view presumably. If so, then the benefit really comes from lossiness rather than temporal correlation?)

---

### After Rebuttal

Increasing rating based on the authors' clarifications on the source of the gains. Open to further changes based on further review and discussions with other reviewers

---

> ### Author Response · Authors · 2020-11-17
> **Part 2/2: Please read Part 1/2 prior to reading this**
>
>
> ***Lossy-compression-only baseline***
>
> If we decrease lossyness (by using a smaller $\alpha$) we indirectly reduce the correlation ($q$ gets closer to $0.5$). Therefore, it's not straightforward to decouple the compression savings to be able to quantify them and attribute them to (i) lossy compression or (ii) leveraging temporal correlation. We try to understand the additional contribution due to exploitation of temporal correlation in two steps. First, we follow the reviewer’s suggestion to establish a lossy-compression-only baseline, then we consider our scheme.
>
> + Following the reviewer's suggestion to design a baseline, we have come up with the following comparison. Let us consider a new encoder/decoder pair. The encoder computes $\text{sgn}(\hat{g}_k[j])$, for which, according to the left-hand subfigure of Figure 2, the fractions of $+1$ and $-1$'s are equal. The encoder then randomly flips the sign of $+1$'s with probability $\alpha$. The resulting vector has a fraction of $+1$’s equal to $0.5(1-\alpha)$ and therefore can be compressed.  Picking $\alpha = 0.7$, we would compute $H(0.5(1-0.7)) = H(0.15) = 0.61$, i.e., compression of 39%. Not bad.  (Note that $H(\cdot)$ is the binary entropy function of a Bernoulli-$\gamma$ source where $0 \leq \gamma \leq 1$, defined as $H(\gamma) = -\gamma \log_2(\gamma) – (1-\gamma) \log_2 (1-\gamma)$.)
>
> + Next, we consider our scheme. Instead of compressing $\hat{g}$ we apply the same lossy compression as the baseline, but across time to $\hat{g}$ and $\bar{g}$. However, we do not start with an unbiased source. Rather, due to the use of distortion in previous steps we start with a **biased** source where the bias is $q \leq 0.5$. So, now our compression rate is $H(q(1-\alpha)) < H(0.5(1-\alpha))$. For the same $\alpha = 0.7$ we find from the central figure of Figure 2 that $q = 0.2$ (actually somewhat less) yielding a $p = q ( 1-\alpha) = 0.2 (0.3) = 0.06$ (cf. right-hand plot Figure 2). $H(0.06) = 0.33$, i.e., 67% compression. So, by designing our scheme to induce temporal correlation we increased the compression rate from 39% to 67%, almost doubling the compressive gains and effecting an additional reduction in bit-rate of $0.28$ bits per coordinate.
>
> ---
>
> ***Exploring impact of the batch size on correlation***
>
> We thank the reviewer for suggesting assessing the impact of batch size on the correlation. Following the reviewer’s lead, we have done this and included the results in our updated submission Appendix section A.4 (indicated in blue color). In summary, we observe that $q$ slightly increases with the batch size as the stochastic gradients become more and more positively correlated. However, as per our numerical results, this change remains minimal for typical batch sizes we use in training.

---

> > ### Comment · AnonReviewer4 · 2020-11-24
> > **Thank you**
> >
> > I am increasing my rating based on your clarification on the source of the gains.
> >
> > I would encourage you to put the intuition in Part 2/2 of the comment in the main paper to explain the connection between lossy compression and temporal connection better. The more clearly and simply you can explain that connection, the stronger the paper becomes (so perhaps explain it even more simply/explicitly than you have in the comment).
> >
> > I need to think more about what you have said, but open to further changes based on further thinking and discussions with other reviewers.

---

> > > ### Author Response · Authors · 2020-11-25
> > > **Thank you for the increased rating. We have also updated the paper.**
> > >
> > > Following the reviewer's latest response we have incorporated to our paper the reasoning for the connection between lossy compression and temporal correlation. Please see the updated text (in blue) on page 5.

---

> ### Author Response · Authors · 2020-11-17
> **Part 1/2: Addressing concerns regarding the source of rate savings**
>
>
> ***Addressing concerns regarding the source of rate savings***
>
> We thank the reviewer for the comments and the suggestion to implement a baseline. First, we agree that our scheme’s benefits derive from two sources: (i) the use of lossy compression and (ii) the exploitation of temporal correlation.  While it would be ideal to be able to decouple the respective contributions from each, it is a bit difficult to do so as we have explicitly designed the lossy compressor ALSO to introduce temporal contribution. We will do our best to describe the relative importance of each of (i) and (ii), and the coupled impact of exploiting both effects. Also, we have updated our manuscript (second paragraph on Page 5 and Section 5.2) to better reflect this reasoning. The updated text is indicated in blue color.
>
> + First, as the reviewer noted, Figure 2 is the right place to look to see what is going on. The left-hand figure plots $r$, the marginal distribution of $+1$’s in $\text{sgn}(\hat{g}_k)$ for different levels of distortion.  $\alpha = 0$ is no distortion while $\alpha = 0.95$ is a lot of distortion. We first observe that the marginal distribution remains unchanged for different $\alpha$. In other words, when distortion is applied in the **previous** iteration, the compression gain in the **current** iteration is negligible if we try to compress $\text{sgn}(\hat{g}_k)$ on its own (since $r\approx0.5$ and $H(0.5)=1$).
>
> + Second, the measure of temporal correlation is $q$. This is plotted in the central sub-figure of Figure 2 and quantifies the temporal correlation between $\hat{g}_k$ and $ \bar{g}_k$. Specifically, $q$ is the fraction of entries such that $\text{sgn}(\hat{g}_k[j]) = \text{sgn}(\bar{g}_k[j])$, where $\hat{g}_k$ (current gradient) and $ \bar{g}_k$ (previous gradient) are the inputs to the encoder. $q$ is therefore a measurement of the correlation that is naturally present between $\hat{g}_k$ and $ \bar{g}_k$ (before introducing any further distortion). The more different $q$ is from $0.5$ the greater the temporal correlation (positive or negative correlation). IMPORTANTLY note that $q$ is measured **PRIOR** to adding distortion through lossy compression using the $\alpha$ parameter. Through this plot we observe that as $\alpha$ is increased (again larger $\alpha$ corresponds to more compression, i.e., lower bit-rate and higher distortion) the temporal correlation also increases.
>
> + Finally, the third, right-hand sub-figure of Figure 2 plots $p$ which measures the combined effect of using both (i) lossy compression and (ii) exploiting temporal correlation in compression. In particular, $p$ measures the fraction of $1$'s in vector $b$ that encodes the differences of the signs between $\hat{g}_k$ and $ \bar{g}_k$. Without any distortion, the difference vector starts at composition $q$ (because of the already existing temporal correlation). After adding distortion according to parameter $\alpha$ (lossy compression), $b$ yields composition $q (1-\alpha)$ yielding a source that is more compressible.
>
> Looking across all three subfigures of Figure 2 one can observe that a larger $\alpha$ (more compression) leads to a smaller $q$ (more temporal correlation) to produce a smaller $p$ (even more compression). Again, we emphasize that $q$ does *not* depend on the errors introduced by lossy compression in the *current* iteration. Rather, $q$ depends on $\alpha$ only through the *past* iterations. In other words, the temporal correlation observed in the current step is *induced* by the lossy compression of previous steps.

---

### Decision · Program_Chairs · 2021-01-08
**Final Decision**

**Decision:**

Reject

**Comment:**

The paper introduces a new scheme for compressing gradients in distributed learning which is argued to exploit temporal correlation.

The paper received very detailed reviews and generated a lot of discussions (thank you to the reviewers for the amazing job).  Many reviewers acknowledge that this is interesting work, a simple and potentially useful algorithm but pointed out many problems with discussion, theoretical analysis, and experiments (e.g., it was not clear to R4 and R3 that these are temporal correlations which are beneficial rather 'lossiness'). Some of these issues were addressed and the current version is currently much stronger than the initial submission (and stronger than the low average scores suggest). Still, the reviewers do not believe that the paper is ready for publication and I share this sentiment. I would strongly encourage the authors to invest more effort in addressing the reviewers' comments and resubmit the work to one of the upcoming top conferences.